# A Clinician’s View of Wernicke-Korsakoff Syndrome

**DOI:** 10.3390/jcm11226755

**Published:** 2022-11-15

**Authors:** Jan W. Wijnia

**Affiliations:** Slingedael Center of Expertise for Korsakoff Syndrome, Slinge 901, 3086 EZ Rotterdam, The Netherlands; j.wijnia@leliezorggroep.nl; Tel.: +31-10-29-31-555

**Keywords:** refeeding syndrome, delirium, Wernicke encephalopathy, Korsakoff syndrome, alcohol withdrawal, acetylcholine, magnesium, seizures, gait ataxia

## Abstract

The purpose of this article is to improve recognition and treatment of Wernicke-Korsakoff syndrome. It is well known that Korsakoff syndrome is a chronic amnesia resulting from unrecognized or undertreated Wernicke encephalopathy and is caused by thiamine (vitamin B1) deficiency. The clinical presentation of thiamine deficiency includes loss of appetite, dizziness, tachycardia, and urinary bladder retention. These symptoms can be attributed to anticholinergic autonomic dysfunction, as well as confusion or delirium, which is part of the classic triad of Wernicke encephalopathy. Severe concomitant infections including sepsis of unknown origin are common during the Wernicke phase. These infections can be prodromal signs of severe thiamine deficiency, as has been shown in select case descriptions which present infections and lactic acidosis. The clinical symptoms of Wernicke delirium commonly arise within a few days before or during hospitalization and may occur as part of a refeeding syndrome. Wernicke encephalopathy is mostly related to alcohol addiction, but can also occur in other conditions, such as bariatric surgery, hyperemesis gravidarum, and anorexia nervosa. Alcohol related Wernicke encephalopathy may be identified by the presence of a delirium in malnourished alcoholic patients who have trouble walking. The onset of non-alcohol-related Wernicke encephalopathy is often characterized by vomiting, weight loss, and symptoms such as visual complaints due to optic neuropathy in thiamine deficiency. Regarding thiamine therapy, patients with hypomagnesemia may fail to respond to thiamine. This may especially be the case in the context of alcohol withdrawal or in adverse side effects of proton pump inhibitors combined with diuretics. Clinician awareness of the clinical significance of Wernicke delirium, urinary bladder retention, comorbid infections, refeeding syndrome, and hypomagnesemia may contribute to the recognition and treatment of the Wernicke-Korsakoff syndrome.

## 1. Introduction

It is well known that Korsakoff syndrome is a chronic amnesia resulting from unrecognized or undertreated Wernicke encephalopathy and is caused by thiamine (vitamin B1) deficiency. The combination of Wernicke encephalopathy and Korsakoff syndrome is also called Wernicke-Korsakoff syndrome. The Wernicke-Korsakoff syndrome is mostly related to alcohol addiction, but can also occur in other thiamine deficiency related conditions. For example, in patients who underwent bariatric surgery or patients with hyperemesis gravidarum, anorexia nervosa, chronic inflammation in Crohn disease, or ulcerative colitis [1]. Other less well-known causes are HIV, SARS-CoV-2, and other viral infections [2,3,4,5], bacterial infections [6], end-stage cancer in palliative care [7], and hunger strike [8]. Although the diagnosis and treatment of patients with Wernicke-Korsakoff syndrome has been described for decades based on studies of biochemical processes [9,10], clinical presentation [10,11], pathological-anatomical [12] and neuroimaging findings [13], daily practice still shows challenges in diagnosing and treating these patients. The goal of this article is to provide a clinical framework for (timely) recognition and adequate treatment of thiamine deficiency in the acute phase of Wernicke encephalopathy to prevent permanent brain damage associated with Korsakoff syndrome. This article elaborates on existing knowledge and provides succinct information to be used as a diagnostic guide for clinicians who provide initial care for Wernicke-Korsakoff patients, irrespective of the medical specialty. 

## 2. Wernicke Encephalopathy

Wernicke encephalopathy may be identified by the presence of a delirium in malnourished alcoholic patients who have trouble walking. In these patients, the delirium is often caused by thiamine deficiency or in combination with thiamine deficiency (Table 1), which may be erroneously diagnosed as alcohol withdrawal delirium [14]. 

The clinical presentation of thiamine deficiency includes loss of appetite, dizziness, tachycardia, and urinary bladder retention that can be attributed to anticholinergic autonomic dysfunction, as well as confusion or delirium [15,16], which is part of the classic triad of Wernicke encephalopathy [17]. These triad signs of Wernicke encephalopathy are described as ocular motility abnormalities: external ophthalmoplegia and/or nystagmus, ataxia affecting primarily the gait, and confusion or delirium [11]. Caine and colleagues proposed four specific criteria for the clinical identification of Wernicke encephalopathy which consist of (i) the presence of dietary deficiencies, (ii) oculomotor abnormalities, (iii) cerebellar dysfunction, and (iv) either an altered mental state or mild memory impairment [18]. Using any two of Caine’s criteria would greatly improve diagnostic sensitivity from 9 out of 40 patients (23%) to 34 out of 40 (85%). Thiamine deficiency is also known to carry a risk of infections, such as pneumonia, urinary tract infections, abscesses, empyema, and sepsis with or without a known source [6]. In the initial Wernicke phase, infections were reported in 35/68 (51%) patients [6]. These infections can be the heralding sign of severe thiamine deficiency, which has been shown in select case descriptions presenting with infections and lactic acidosis [19,20]. An infection can increase the body’s use of thiamine and may precipitate Wernicke encephalopathy in patients with marginal thiamine reserves [21,22]. Thus, in malnourished patients, infections can be a presenting symptom as well as a complicating factor of thiamine deficiency. 

## 3. Korsakoff Syndrome

The diagnosis of Korsakoff syndrome can be made according to the DSM-5 criteria for major neurocognitive disorder of the confabulating amnestic type [23]. In general, Korsakoff syndrome is characterized by severe anterograde and, to a lesser extent, retrograde amnesia for declarative knowledge [11]. Patients with Korsakoff syndrome may also have difficulty in correctly identifying the temporal sequence of events [24]. Moreover, many patients have executive function deficits, such as problems with initiating, planning, organizing, and regulating behavior [25]. Patients themselves often do not recognize their problems in daily functioning because of limited awareness of their illness (anosognosia). Patients with Korsakoff syndrome can exhibit confabulations, although the intensity and frequency can vary per patient [26]. Furthermore, Korsakoff syndrome is very often accompanied by a peripheral neuropathy [27].

## 4. Frontal Dysfunction in Korsakoff Syndrome

Although the mechanisms of the cognitive dysfunction are still not fully understood, loss of function in the Papez and frontocerebellar circuits [28,29] both including parts of the thalamus [30], may cause the impaired memory and executive functions that are the main characteristics of Korsakoff syndrome [11,31]. Some patients with Korsakoff syndrome suffer from additional damage in the cerebellum [13]. In our experience, this is accompanied by more serious disturbances in regulating behavior, such as perseveration and rigidity, and can increase caregiver burden. 

## 5. Pathophysiology 

Early changes of astrocytes and microglia have been reported in experimental thiamine deficiency [32] and Wernicke encephalopathy [33]. In thiamine deficiency, the earliest biochemical change is the decrease of α-ketoglutarate-dehydrogenase activity (α-KGDH) in astrocytes [10]. Astrocyte dysfunction in thiamine deficiency involves a loss of glutamate transporters and other astrocyte-specific proteins which together contribute to focal neuronal injury in terms of neural cell excitotoxicity caused by extracellular build-up of glutamate. A reduction in the thiamine-dependent activity of transketolase leads to a lower use of glucose and oxidative stress secondary to endothelial cell dysfunction. This produces cytotoxic and vasogenic edema firstly in astrocytes, then in neurons along with disruption of the blood–brain barrier [10,34] and local petechial hemorrhages [35] in brain areas that are specifically vulnerable to thiamine deficiency [10,36,37,38]. Subsequently, neuronal DNA fragmentation and lactic acidosis occur in astrocytes and neurons, leading to necrosis and irreversible structural damage [10]. Other studies have focused on inflammation with microglia hyperactivity and pro-inflammatory cytokines in the cellular response to thiamine deficiency [39,40], which might be an explanation for the focal neuronal loss of this disorder [32]. The function of microglia is normally protective, but defensive features can turn neurotoxic and cause neuronal injury through ongoing microglial overstimulation once microglia cells are no longer inhibited by cholinergic neurons, as described by Van Gool and colleagues [16]. From a pathophysiological point of view, thiamine seems to be involved in acetylcholinergic synaptic transmission [10,41], as thiamine deficiency may cause decreased bioavailability of acetylcholine [42]. This may be due to low acetylation rates in acetylcholine production or to selective vulnerability of cholinergic neurons (Figure 1), but the exact underlying mechanisms remain unclear [43,44].

## 6. Neuroimaging

In magnetic resonance imaging (MRI) during the early Wernicke phase of the disorder, Sullivan and Pfefferbaum (2009) have shown an altered signal in various components of the limbic, circuits including the paraventricular regions of the thalamus, the hypothalamus, mammillary bodies, the periaqueductal region, the floor of the fourth ventricle and midline cerebellum [13,31]. The sensitivity of MRI in detecting Wernicke encephalopathy is only 53%, with a specificity of 93% [13]. On diffusion tensor imaging (DTI), Segobin and colleagues (2019) found that the anterior thalamic nuclei were mainly connected with the hippocampi (84% of parcellations) with a significantly reduced connectivity in Korsakoff patients. The medial dorsal nuclei were mainly connected with frontal-executive brain regions (70% of parcellations) with reductions in both Korsakoff and non-Korsakoff alcoholics, which was associated with atrophy of mediodorsal nuclei [30]. On FDG-PET (18-fluoro-deoxy-glucose positron emission tomography), Reed and colleagues (2003) showed metabolic changes in the thalami and mammillary bodies, and also in surrounding tissue, namely the hypothalamus, a small portion of the basal forebrain, and the retrosplenium, all components of the anterograde memory (limbic) circuitry [31,46].

## 7. Prognosis 

In the initial Wernicke phase, twenty-five of 468 patients (5.3%) with Wernicke encephalopathy died during hospitalization. The causes of death were cancer, cardiac arrest, infections, and head injury [47]. In a tertiary hospital study of Sanvisens and colleagues, the median survival time in 51 patients with Wernicke encephalopathy was 8.0 years (95%-CI: 5.3–10.7 years) and mortality was associated with infection, cancer, or malnutrition. Two-thirds of the patients continued alcohol use after discharge, of whom 6% presented with a subsequent Wernicke episode [48]. In a national population-based register study by Palm and colleagues (2022), the median survival in Wernicke-Korsakoff syndrome and alcohol-related dementia was 10.7 years (95%-CI: 9.6–11.3 years) and 5.9 years (95%-CI: 5.6–6.3 years), respectively [49]. The main causes of death of people with a diagnosis of Wernicke-Korsakoff syndrome were diseases of the circulatory system (24.0%), neoplasms (16.4%), diseases of the digestive system (16.0%), mental and behavioral disorders (13.3%), and accidents, suicides and other external causes (12.1%) [49]. The main causes of death in 138/349 (39.5%) patients receiving residential Korsakoff care were cancer (in 40.6%), infections (26.8%), and sudden death (8.0%). The median survival was 5.8 years (IQR: 3.0–8.9 years) after the initial hospitalization because of Wernicke encephalopathy [50]. 

## 8. Alcoholic versus Non-Alcoholic Wernicke Encephalopathy

Wernicke encephalopathy is mostly related to alcohol addiction (400/434 cases = 92.2%). The mean age of these patients with Wernicke encephalopathy was 55.1 (SD 11.8) years and 50.7 (SD 16.5) years, respectively in alcohol related and non-alcohol related cases, and the gender ratio (male/female) was, respectively 6.0 and 0.7 [47]. Most patients with alcohol related Wernicke-Korsakoff syndrome (103/128 = 80.5%) are initially admitted following a state of confusion with impaired consciousness, or after a physical collapse [6,31]. Some patients with alcohol related Korsakoff syndrome (9/118 = 8%) show intermittent episodes of Wernicke encephalopathy over time [6]. Alcoholic patients presented more frequently (368/434 = 84.8%) than non-alcoholic patients (23/34 = 68%) with cerebellar signs, but less frequently with ocular signs (65.7% versus 85%). Alcoholic patients had a significantly higher frequency of hyponatremia compared with non-alcoholics (105/425 = 24.7%, respectively 9% of patients) and lower platelet counts (mean 227 × 10^3^/μL, respectively 281 × 10^3^/μL). The median time from hospital admission to Wernicke encephalopathy diagnosis was 1 versus 4 days, respectively, in alcoholic and non-alcoholic patients [47]. Vomiting (81.9% of patients) and weight loss (mean 18.3 kg weight loss) commonly characterized the onset of non-alcohol-related Wernicke encephalopathy [1]. Complaints of blurred vision were reported in 24.3% of Wernicke cases in hyperemesis gravidarum and may be associated with optic neuropathy in thiamine deficiency [1]. Cognitive deficits may be less prominent in non-alcoholic Wernicke encephalopathy, suggesting relatively lower susceptibility to confusion in this patient population. This is especially the case among younger patient groups or patients who seek treatment earlier than those struggling with alcoholism. For instance, in thiamine deficiency after bariatric surgery, patients presenting without mental status changes were on average 11 years younger than those with mental status changes (Mann–Whitney U test, U (66) = 262, *p* < 0.005) [51].

## 9. A Missed Diagnosis of Wernicke Encephalopathy

The acute Wernicke encephalopathy in thiamine deficiency presents as delirium, but often goes unnoticed for several reasons. First, specialization in medical fields may contribute to a lack of overview in thiamine deficiency and its manifestations, which do not adhere to DSM-5 or ICD-11 classifications. Apart from the American psychiatric association that lists thiamine deficiency among the underlying metabolic disorders commonly associated with delirium [52], most guidelines on delirium do not mention thiamine deficiency among the relevant causes of delirium [53,54,55,56]. In guidelines on alcohol use disorders, delirium is mainly mentioned in relation to alcohol withdrawal [57,58,59]. Secondly, a delirium of any cause is recognized in only 30% to 70% of cases [60]. In daily practice, if a delirium diagnosis is made in patients with alcohol abuse, the delirium is often labeled as an alcohol withdrawal delirium, thus missing the Wernicke delirium that may present with similar symptoms of attention deficits, slurred speech, impaired consciousness, and loss of ability to walk. Furthermore, in patients suffering from Wernicke encephalopathy, concomitant infections can cause severe complications that further mask a relevant cause of their impaired consciousness. In these critically ill patients, a full neurological examination of ocular motility abnormalities and uncoordinated movements of the legs may be difficult to accomplish. However, repeated examination during follow-up may show disturbances of smooth eye pursuit, abnormal heel-to-shin testing, and ultimately a wide based ataxic gait without inebriation [18].

## 10. Manifestations of Wernicke Delirium

Like any delirium, the delirium in Wernicke encephalopathy results in a reduced awareness of the environment, with impaired ability to focus and sustain attention (decreased tenacity), and sometimes increased distractibility (hypervigilance). Drawing from the author’s own personal experience, if the diagnostic interview is paused for 30 s, a decrease in the patient’s attention will be noticed. In medical literature, little attention is paid to specific motor characteristics of delirium in general, notably the presence of dysarthria resulting in slurred speech in delirium [61]. When a delirious patient answers a question, the sentences may begin well-articulated and then become unintelligible, presumably because of an exhaustion effect in neurotransmitters such as acetylcholine [62]. This phenomenon may repeat itself with every new sentence. In addition to these motor signs of uncoordinated speech, Wernicke delirium can result in unsteady walking or wheelchair dependency. 

## 11. The Course of Illness

The Wernicke-Korsakoff syndrome is a prototype of delirium progressing to cognitive deficits, if the initial encephalopathic stage has resulted in lasting damage (Figure 2).

Regarding the mobility disturbances and motor signs of ataxia in Wernicke-Korsakoff syndrome, it takes some months before a plateau phase is reached corresponding with a chronic Korsakoff stage (Figure 3). 

A Korsakoff diagnosis is often suspected in the first weeks of hospital admission, but usually this is still a Wernicke delirium that is partially in remission. The neuropsychological assessment of Korsakoff syndrome can start after at least six weeks of alcohol abstinence [6]. This period may be even longer if delirium features persist. Occasionally, the Wernicke phase and attention deficits remain chronically present. These patients appear to be permanently wheelchair dependent (Figure 2) and often function on a dementia level which is lower than the cognitive level usually found in Korsakoff syndrome. Most strikingly are the I-don’t-know answers (agnosia) in this patient group, whereas patients with Korsakoff syndrome are more likely to fill in their memory gaps. However, even after severe thiamine deficiency, it can occasionally take more than a year before the definite stage of functioning has been reached [64]. In all cases, a timely and adequate treatment in the acute phase is crucial.

## 12. Comorbidity

In patients with alcohol-related Wernicke-Korsakoff syndrome, depression was reported in 18% of the patients, psychotic disorders and hallucinosis in 7%, personality disorders in 16%, and other psychiatric disorders in 7% of the patients [65]. Psychiatric conditions such as schizophrenia, pervasive developmental disorder, and schizotypal personality disorders may have been missed until the patient receives medical attention due to the consequences of vitamin deficiencies. Somatic comorbidities are a serious problem in patients with Wernicke-Korsakoff syndrome, and include among others chronic obstructive pulmonary disease (in 34.4%), liver cirrhosis (26.9%), peripheral arterial disease (22.9%), stroke (22.6%), diabetes mellitus (21.8%), epilepsy (13.2%), myocardial infarction (12.3%), and Barrett esophagus (2.3%), according to data of Wernicke-Korsakoff patients in residential care [50]. In a study of Novo-Veleiro and colleagues (2022), patients with alcohol related Wernicke encephalopathy and liver disease presented more frequently with tremor (26.5% versus 14.0%), flapping (10.5% versus 3.3%), and hallucinations (35.2% versus 21.3%) than those without alcoholic liver disease [66]. Of the patients with suspected Korsakoff syndrome, who were admitted to our diagnostic center, new malignancies were diagnosed in 87/389 (22.4%) patients, including tumors of mouth, throat and esophagus, lung cancer, and colon cancer, median 3 years after admission [50]. Importantly, pain perception in Korsakoff syndrome may be disturbed, probably due to higher pain thresholds [67]. Consequently, the alerting signals of pain may be less well perceived by Korsakoff patients, which may cause a delay in help-seeking behavior.

## 13. Refeeding Syndrome

Whenever Wernicke encephalopathy is suspected, treatment should be initiated immediately [68]. Symptoms of Wernicke encephalopathy, such as drowsiness, confusion, and walking disability leading to collapse, develop on average within 3 to 4 days before a subsequent hospital admission [6]. However, Wernicke encephalopathy may be precipitated by a refeeding syndrome within 2 to 3 days in a hospital [69], as most thiamine deficient patients have not been eating properly for months and sometimes have not eaten at all for several days or weeks [65]. These patients should be gradually reintroduced to caloric intake in consultation with a dietitian. They should also be carefully monitored during the first days of admission, including serial electrolyte and blood glucose checks [70]. The refeeding syndrome is characterized by water-electrolyte imbalance, in particular hypophosphatemia, hypokalemia and hypomagnesemia, glucose intolerance, manifestation of thiamine deficiency, and fluid overload [71]. Reintroducing carbohydrates or administering glucose without the addition of thiamine can be a risk factor for the development of Wernicke encephalopathy in already depleted thiamine stores [72]. Criteria for a high risk of developing refeeding syndrome are found in the American Society for Parenteral and Enteral Nutrition (ASPEN) Consensus Recommendations for Refeeding Syndrome [73]. If there are no abnormal laboratory values for phosphate, calcium, potassium, magnesium and glucose (glucose curves), monitoring in the context of refeeding syndrome is stopped after 72 h [74]. Several other risk factors may significantly contribute to thiamine deficiency, including infections, esophageal stenosis (Barrett esophagus), colitis, and importantly renal loss of thiamine in diabetes mellitus or nephropathy [75,76]. Loss of appetite and vomiting may be both a cause [77] and a complication of thiamine deficiency [78,79].

## 14. Thiamine Treatment

Regarding the treatment of Wernicke encephalopathy, there is a clear controversy about the doses that should be administered both in prophylaxis and treatment. Despite this, it should be stressed that rapid administration of thiamine is important. The problem lies in choosing the most appropriate dose. Regarding prophylaxis, its recommendation is based on patients undergoing bariatric surgery. Additionally, despite the scant evidence and the fact that the guidelines do not include it, prophylaxis should be performed in hospitalized patients with alcohol use disorders [80,81]. If there is a high risk of Wernicke encephalopathy, 250 mg of parenteral (intravenous of intramuscular) thiamine is advised, once a day for 3 to 5 days [82] even though different recommendations and limited data exist in this area [9,70]. For adequate treatment of Wernicke encephalopathy, the recommended dose is 200 mg three times per day according to the EFNS guidelines [83]. Seven out of nine other guidelines [80,82,83] recommend >500 mg parenteral thiamine per day. Four of these guidelines [80,82] recommend administering of intravenous thiamine, slowly over 30 min and diluted in 100 mL of normal saline [82], three times a day for 3 to 5 days [80], followed by 250 mg intravenous daily for a minimum of 3 to 5 additional days [84] or followed by oral doses [80]. In extreme cases, if a voluntary or compulsory emergency hospital admission cannot be realized and intravenous dosing is not feasible, the patient with suspected Wernicke encephalopathy can be treated with 250 mg of intramuscular thiamine per day [85]. In this case, oral thiamine supplementation is entirely ineffective in preventing permanent brain damage [68,85]. Furthermore, blood draw should not lead to any delay in treatment [68]. Importantly, initial serum thiamine concentrations can be normal despite the clinical signs of Wernicke encephalopathy. For instance if hypomagnesemia is present, patients may fail to respond to parenteral thiamine [86,87]. 

## 15. Hypomagnesemia

Thiamine is dependent on magnesium for its role in metabolizing glucose in the energy generating processes of the pentose phosphate pathway and the Krebs cycle in the mitochondria of the cells. Consequently, thiamine supplementation may be ineffective if existing or developing magnesium deficiencies are not corrected at the same time. Seizures may occur during both Wernicke encephalopathy [88] and alcohol withdrawal [89], and are also associated with magnesium deficiency [90]. In patients with alcohol withdrawal symptoms, most showed low serum magnesium concentrations and were still under-supplemented during follow-up appointments [89]. The infrequency of magnesium supplementation in the context of low serum magnesium concentrations may reflect a lack of clinician awareness regarding the clinical significance and prevalence of magnesium deficiency in patients with alcohol use [89]. Furthermore, apart from alcohol abuse, malnutrition, or refeeding syndrome, hypomagnesemia may exist due to adverse side effects of proton pump inhibitors and whether or not it is combined with diuretics. For instance, in somatic comorbidity of alcohol-related esophageal peptic ulcers and ascites due to liver cirrhosis. 

## 16. Summary and Conclusions

It is well known that Korsakoff syndrome is a chronic amnesia resulting from unrecognized or undertreated Wernicke encephalopathy and is caused by thiamine (vitamin B1) deficiency. The clinical presentation of thiamine deficiency includes loss of appetite, dizziness, tachycardia, and urinary bladder retention that can be attributed to anticholinergic autonomic dysfunction, as well as confusion or delirium, which is part of the classic triad of Wernicke encephalopathy. The clinical symptoms of Wernicke delirium commonly arise within a few days before or during hospitalization, sometimes as part of a refeeding syndrome in malnourished patients. Wernicke encephalopathy may be identified by the presence of delirium in malnourished alcoholic patients who have trouble walking. The onset of non-alcohol-related Wernicke encephalopathy is more commonly characterized by vomiting, weight loss, and symptoms such as visual complaints due to optic neuropathy, which may occur due to thiamine deficiency. Importantly, patients may fail to respond to parenteral thiamine in the presence of hypomagnesemia. Severe infections including sepsis of unknown origin can be both a precipitating factor as well as a result of thiamine deficiency. We hope an interdisciplinary approach may further improve the diagnosis and timely, adequate treatment of patients at risk of or suffering from Wernicke-Korsakoff syndrome.

## Figures and Tables

**Figure 1 jcm-11-06755-f001:**
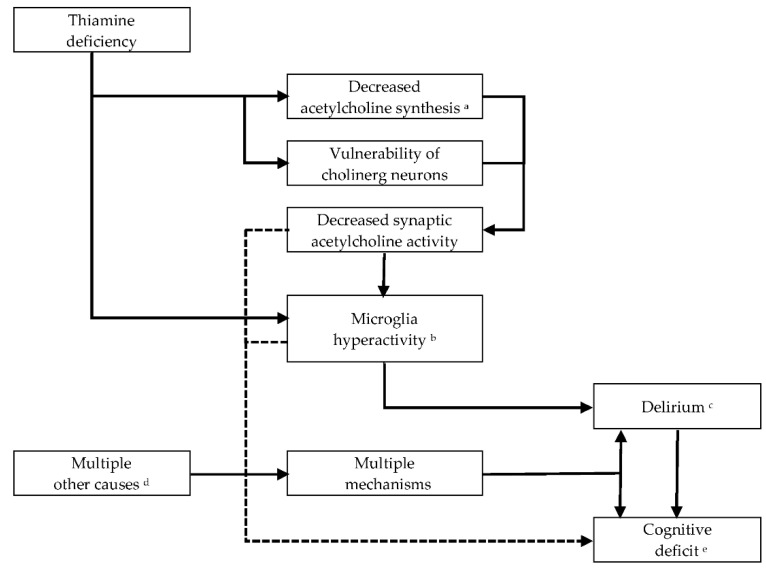
Acetylcholine in Wernicke-Korsakoff syndrome. ^a^ Acetylcholine synthesis: Choline is acetylated to the neurotransmitter acetylcholine by choline acetyltransferase (CAT) and by use of acetyl-CoA. In the metabolism of glucose, the enzymatic step that is catalyzed by pyruvate dehydrogenase requires thiamine pyrophosphate, the active form of thiamine. Reduced activity of pyruvate dehydrogenase may result in deficiency of acetyl-CoA and reduced synthesis of acetylcholine [45]. ^b^ Wang and Hazell (2011) observed induction of microglia activity in their thiamine deficient rodent model [39]. ^c^ Based on an hypothesis of van Gool and colleagues (2011) microglia hyperactivity may lead to a delirium [16]. Delirium due to thiamine deficiency = Wernicke encephalopathy [15]. ^d^ For instance, alcohol withdrawal, brain injury, hepatic failure, renal failure, brain injury, infections, and electrolyte disorders. ^e^ Neurocognitive disorder due to thiamine deficiency = Korsakoff syndrome.

**Figure 2 jcm-11-06755-f002:**
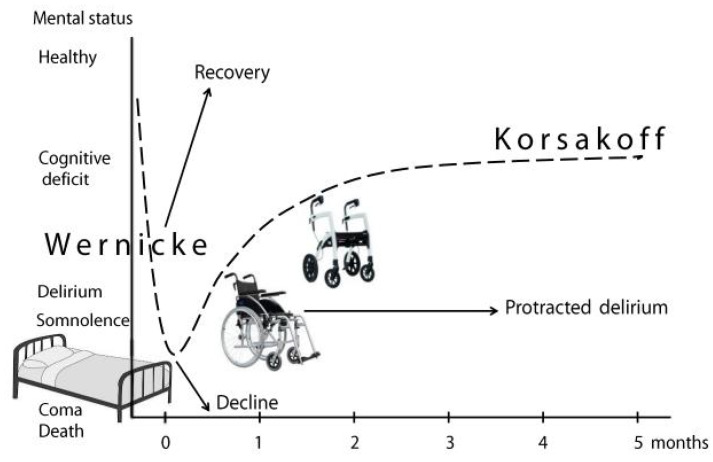
Mental and motor symptoms in Wernicke-Korsakoff syndrome. A patient’s mobility and evolving neuropsychiatric symptoms in Wernicke-Korsakoff syndrome (dashed line), starting with initial hospitalization at 0 months. Arrows depict alternative outcomes: further decline, full recovery, or protracted delirium.

**Figure 3 jcm-11-06755-f003:**
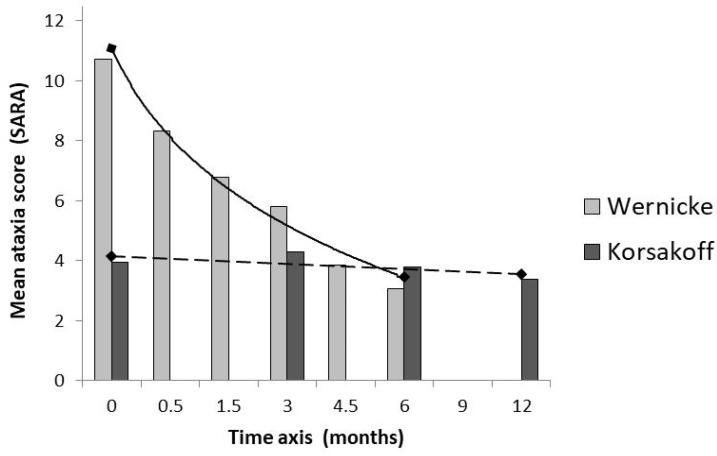
Ataxia in Wernicke-Korsakoff syndrome. SARA, the Scale for Assessment and Rating of Ataxia is an 8-item performance-based scale, yielding a total score of 0 = no ataxia to 40 = most severe ataxia [63]. Bars represent mean sum scores of fifteen patients in the Wernicke phase and ten other patients in the chronic Korsakoff phase. The time axis shows the follow-up of SARA scores in both patient groups and starts, respectively on the date of admission in the Wernicke group and on the date of the first measurements in Korsakoff patients, who have been admitted for 18 months or more, in Slingedael Korsakoff center, Rotterdam.

**Table 1 jcm-11-06755-t001:** Differential diagnosis of confusion, agitation, or drowsiness in patients with alcohol use disorder admitted to our center because of suspected Wernicke-Korsakoff syndrome.

**Complications of alcohol abuse**
Alcohol intoxication Alcohol withdrawal: delirium tremens, seizures Hypoglycemia, acidosis ^a^ Electrolyte disorders: hyponatremia, hypomagnesemia
**Acute or progressive cerebral disorders**
Wernicke encephalopathy ^b^ Osmotic demyelination syndrome Head injury: subdural hematoma, concussion Stroke Space-occupying lesion of the brain Hepatic encephalopathy Alcoholic cerebellar degeneration Marchiafava-Bignami disease ^b^
**Severe infections ^a^**
Pneumonia, urinary tract infection, urosepsis, sepsis of unknown origin, abscesses, empyema, meningitis, endocarditis, spondylodiscitis, other infections
**Acute abdominal conditions**
Gastrointestinal bleeding: ulcers, esophageal varices Portal hypertension, alcoholic hepatitis, liver cirrhosis Alcoholic pancreatitis Peritonitis associated with ascites Acute mesenteric ischemia, other vascular diseases
**Decompensation in response to other conditions**
Dysregulation of diabetes mellitus, dehydration, nephropathy (renal loss of thiamine) Myocardial infarction, heart failure, cardiomyopathy ^a^ Severe chronic obstructive pulmonary disease, emphysema, hypoxia, hypercapnia ^c^ Malignancy: mouth, throat, lung ^c^ Esophagus stenosis (Barrett, carcinoma)
**Delirious state**
May accompany all causes listed above
**Other neuropsychiatric conditions**
Korsakoff syndrome ^b^ Dementia Psychosis, schizophrenia Personality disorders, e.g., schizotypal, paranoid Depression and anxiety disorders
**Other substance dependences or intoxications**
Illicit drugs, benzodiazepines

^a^ Possibly caused by or in combination with thiamine deficiency. ^b^ Caused by thiamine deficiency. ^c^ In combination with smoking. Conditions were collected from medical records of patients admitted to Slingedael Korsakoff center, Rotterdam, The Netherlands.

## Data Availability

Not applicable.

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
