# Peer review of "A Clinician’s View of Wernicke-Korsakoff Syndrome"

_jcm, 2022, doi:10.3390/jcm11226755_

Round 1
Reviewer 1 Report
The authors review Wernicke's encephalopathy (WE) from a clinical point of view. However, I believe that several points should be resolved before being accepted in this journal.
Major comments:
-From a pathophysiological point of view, the mechanism that is exposed to the production of acetylcholine does not justify the hypoxic-ischemic lesions that appear in necropsies. I propose to review the decrease of the activity of alpha-ketoglutarate-dehydrogenase, which leads to a lower use of glucose and oxidative stress secondary to endothelial cell dysfunction; this produces cytotoxic and vasogenic edema in astrocytes and neurons along with disruption of the blood-brain barrier. Subsequently, neuronal DNA fragmentation and lactic acidosis occur in astrocytes and neurons, leading to necrosis and irreversible structural damage to specific brain areas. Kohnke S, et al. Ann Clin Biochem. 2021;58:38–46. Sechi G, et al. Lancet Neurol. 2007;6:442–55.
-In point 4 there is no mention of the differences between alcoholic and non-alcoholic patients from a clinical, analytical and prognostic significance of view described in the paper of Chamorro et al (10.1016/j.mayocp.2017.02.019) with 468 patients with WE and that I consider being relevant as it is one of the most significant observational studies on WE described to date.
-I consider that the exposition of a clinical case, as it appears at the end of points 6 and points 9, is irrelevant in this review.
-Among the comorbidities in point 9, mention should be made of the differences described between patients with WE with liver disease and those without liver disease described by Novo-Veleiro et al (DOI: 10.1016/j.drugalcdep.2021.109186.).
-It is surprising that no mention is made of the radiological diagnosis in the exposition of the review. This is a point that should not be left out of any review of Wernicke's encephalopathy. I believe that with the elimination of the exposed clinical cases there would be enough space to create a point of diagnosis through complementary tests.
-Regarding treatment, there is a clear controversy about the doses that should be administered both, in prophylaxis and treatment. According to the EFNS guidelines (Galvin et al DOI: 10.1111/j.1468-1331.2010.03153), the recommended dose for treatment is 200mg three times per day. Despite this, it should be stressed that what is important is the rapid administration of thiamine and the difficulty of establishing clinical trials to verify which is the most appropriate dose. Regarding prophylaxis, its recommendation has been established in patients undergoing bariatric surgery. And despite the scant evidence and the fact that the guidelines do not include it, prophylaxis should be performed in patients with AUD who are hospitalized, according to different authors Pruckner N, et al. Eur Addict Res. 2019;25:103–10. Praharaj SK, et. to the. Indian J Psychiatry. 2021;63:121–6.
Minor comments:
- Within the introduction, reference should be made to the recent cases of WE associated with SARS CoV2 infection that have been described in the literature.
- Marchiafava-Bignami disease and alcoholic cerebellar degeneration should be included in the differential diagnosis table. The old term central pontine myelinolysis should be replaced by osmotic demyelination syndrome, more widely accepted in recent years.
-In point 5 there are no bibliographical references at the end of the text on which to base these statements. The same thing happens in the last paragraph of point 7.
Author Response
Author's Reply to the Review Report of Reviewer 1.
The author reviews Wernicke's encephalopathy (WE) from a clinical point of view. However, I believe that several points should be resolved before being accepted in this journal.
Major comments:
-From a pathophysiological point of view, the mechanism that is exposed to the production of acetylcholine does not justify the hypoxic-ischemic lesions that appear in necropsies. I propose to review the decrease of the activity of alpha-ketoglutarate-dehydrogenase, which leads to a lower use of glucose and oxidative stress secondary to endothelial cell dysfunction; this produces cytotoxic and vasogenic edema in astrocytes and neurons along with disruption of the blood-brain barrier. Subsequently, neuronal DNA fragmentation and lactic acidosis occur in astrocytes and neurons, leading to necrosis and irreversible structural damage to specific brain areas. Kohnke S, et al. Ann Clin Biochem. 2021;58:38–46. Sechi G, et al. Lancet Neurol. 2007;6:442–55.
Answer: We thank the reviewer for these comments for improving the quality and readability of the article. => A review describing the decrease of α-KGDH activity leading to necrosis is added in a new paragraph on pathophysiology, including references. In the second half of this paragraph and in a new Figure (Fig.1), the concept of decreased acetylcholine is further described in relation to microglia hyperactivity in thiamine deficiency.
-In point 4 there is no mention of the differences between alcoholic and non-alcoholic patients from a clinical, analytical and prognostic significance of view described in the paper of Chamorro et al (10.1016/j.mayocp.2017.02.019) with 468 patients with WE and that I consider being relevant as it is one of the most significant observational studies on WE described to date.
Answer: => The clinical and analytical information of Chamorro and colleagues is added in the manuscript. The prognosis of patients with Wernicke Korsakoff is described in a new paragraph (7. Prognosis).
-I consider that the exposition of a clinical case, as it appears at the end of points 6 and points 9, is irrelevant in this review.
Answer: The clinical cases are now omitted.
-Among the comorbidities in point 9, mention should be made of the differences described between patients with WE with liver disease and those without liver disease described by Novo-Veleiro et al (DOI: 10.1016/j.drugalcdep.2021.109186.).
Answer: => This information is now added, including reference.
-It is surprising that no mention is made of the radiological diagnosis in the exposition of the review. This is a point that should not be left out of any review of Wernicke's encephalopathy. I believe that with the elimination of the exposed clinical cases there would be enough space to create a point of diagnosis through complementary tests.
Answer: A new paragraph (=> 6. Neuroimaging) is added.
-Regarding treatment, there is a clear controversy about the doses that should be administered both, in prophylaxis and treatment. According to the EFNS guidelines (Galvin et al DOI: 10.1111/j.1468-1331.2010.03153), the recommended dose for treatment is 200mg three times per day. Despite this, it should be stressed that what is important is the rapid administration of thiamine and the difficulty of establishing clinical trials to verify which is the most appropriate dose. Regarding prophylaxis, its recommendation has been established in patients undergoing bariatric surgery. And despite the scant evidence and the fact that the guidelines do not include it, prophylaxis should be performed in patients with AUD who are hospitalized, according to different authors Pruckner N, et al. Eur Addict Res. 2019;25:103–10. Praharaj SK, et. to the. Indian J Psychiatry. 2021;63:121–6.
Answer: => Above text is added in the paragraph on thiamine treatment.
Minor comments:
- Within the introduction, reference should be made to the recent cases of WE associated with SARS CoV2 infection that have been described in the literature.
Answer: => SARS-CoV-2 is added.
- Marchiafava-Bignami disease and alcoholic cerebellar degeneration should be included in the differential diagnosis table. The old term central pontine myelinolysis should be replaced by osmotic demyelination syndrome, more widely accepted in recent years.
Answer: => ‘Marchiafava-Bignami disease’ is added. ‘Central pontine myelinolysis’ is replaced by ‘Osmotic demyelination syndrome’.
-In point 5 there are no bibliographical references at the end of the text on which to base these statements. The same thing happens in the last paragraph of point 7.
Answer: => References, sources, have now been added.
Reviewer 2 Report
In this manuscript, the author provides a nicely done, well-timed review of Wernicke-Korsakoff syndrome: its etiology, incidence, pathophysiology, clinical features, and treatments. Given the importance of this topic, this is a well-suited manuscript for the readership given a few minor improvements and clarifications:
1. The abstract has some grammatical errors and would be clearer with a careful review. Furthermore, the abstract is not organized in a manner that flows and provides the material in an easily digestible manner.
2. This section (Number 2) on Korsakoff syndrome would be better suited after the description of Wernicke encephalopathy.
3. In section 3: A figure describing the biochemical pathway associated with the development of Wernicke encephalopathy would be helpful. I.e a figure showing the relationship between thiamine, acetylcholine, and acetylcholine acetylation rates
4. The entire article in general (specifically point 4 for example) speaks in very general terms without providing any clear numbers. For example lines 95-96 “Cognitive phenomena however may be less 95 prominent, suggesting relatively lower susceptibility to confusion.”
The manuscript would be much improved by more specific numbers and ranges provided that provide objective data and values for readers to refer to which is key in a review article.
5. The manuscript needs more thorough references along with a more thorough review of the literature given this is a review article. I.e. line 103-104 where “most guidelines” are not cited and it’s not clear where this information was obtained. Additionally, for example, in lines 122-124.
6. The overall article needs to be thoroughly reviewed for grammatical errors that are present throughout the manuscript that hinders readability.
Author Response
Author's Reply to the Review Report of Reviewer 2.
In this manuscript, the author provides a nicely done, well-timed review of Wernicke-Korsakoff syndrome: its etiology, incidence, pathophysiology, clinical features, and treatments. Given the importance of this topic, this is a well-suited manuscript for the readership given a few minor improvements and clarifications:
- The abstract has some grammatical errors and would be clearer with a careful review. Furthermore, the abstract is not organized in a manner that flows and provides the material in an easily digestible manner.
Answer: We thank the reviewer for these comments for improving the quality and readability of the article. => The abstract has been rewritten in a more logical manner.
- This section (Number 2) on Korsakoff syndrome would be better suited after the description of Wernicke encephalopathy.
Answer: => The paragraph on Korsakoff syndrome follows now after the description of Wernicke encephalopathy.
- In section 3: A figure describing the biochemical pathway associated with the development of Wernicke encephalopathy would be helpful. I.e a figure showing the relationship between thiamine, acetylcholine, and acetylcholine acetylation rates.
Answer: => A new Figure (Fig.1) Acetylcholine in Wernicke-Korsakoff syndrome has been added.
- The entire article in general (specifically point 4 for example) speaks in very general terms without providing any clear numbers. For example lines 95-96 “Cognitive phenomena however may be less 95 prominent, suggesting relatively lower susceptibility to confusion.”
The manuscript would be much improved by more specific numbers and ranges provided that provide objective data and values for readers to refer to which is key in a review article.
Answer: => The manuscript now shows more specific numbers and ranges, including references.
- The manuscript needs more thorough references along with a more thorough review of the literature given this is a review article. I.e. line 103-104 where “most guidelines” are not cited and it’s not clear where this information was obtained. Additionally, for example, in lines 122-124.
Answer: => The manuscript now shows more specific information, including references.
- The overall article needs to be thoroughly reviewed for grammatical errors that are present throughout the manuscript that hinders readability.
Answer: I hope readability has improved now.
Round 2
Reviewer 1 Report
I think that the article can be accepted after the changes introduced and that they have improved it significantly